# New C-Glycosidic Ellagitannins Formed upon Oak Wood Toasting, Identification and Sensory Evaluation

**DOI:** 10.3390/foods9101477

**Published:** 2020-10-16

**Authors:** Kleopatra Chira, Laura Anguellu, Gregory Da Costa, Tristan Richard, Eric Pedrot, Michael Jourdes, Pierre-Louis Teissedre

**Affiliations:** 1Department of Environmental Science and College of Grape and Wine Sciences, University of Bordeaux, INRAE, IPB, Œnologie, EA 4577, USC 1366, ISVV, 210 Chemin de Leysotte, F-33140 Villenave d’Ornon, France; laura.anguellu@hotmail.fr (L.A.); gregory.da-costa@u-bordeaux.fr (G.D.C.); tristan.richard@u-bordeaux.fr (T.R.); eric.pedrot@u-bordeaux.fr (E.P.); michael.jourdes@u-bordeaux.fr (M.J.); pierre-louis.teissedre@u-bordeaux.fr (P.-L.T.); 2Tonnellerie Nadalie, 99 rue Lafont—Ludon Medoc—CEDEX, 33 295 Blanquefort, France

**Keywords:** ellagitannins, toasting, oak wood, astringency, bitterness

## Abstract

In the courses of studies on ellagitannin changes during oak wood toasting, two C-glycosidic ellagitannins were isolated from the french oak wood for the first time. These two compounds exhibited [M−H]^−^ ion peak at *m*/*z* 1055.0631 (compound A) and at *m*/*z* 1011.0756 (compound B). A compound is named Castacrenin E and is produced by Castacrenin D oxidation. Castacrenin D is a vescalagin with an additional aromating ring to the C-1 through a C-C bond. These compounds are not only found under laboratory conditions but also in commercial oak wood representing different toasting methods and sizes. Their levels are conditioned by oak wood dimensions and toasting degree. The wood pieces with the smallest size present almost two times more compounds A and B. Moreover, the compound B is the only compound to be present in medium toasting temperatures of the smallest wood pieces. Both of them can influence either astringency sensation or bitterness taste.

## 1. Introduction

Towards the 17th and 18th centuries wine aging techniques using barrels became widespread. The use of many species such as chestnut, acacia, eucalyptus, ash, alder, beech, poplar, pine, and mangrove coexisted with that of oak in the manufacture of barrels. Oak was introduced progressively due to the observation that it is the only wood that allowed to improve wine organoleptic properties [1]. Wine experiences significant changes, such as instinctive clarification, oxygen micro-amounts penetration from oak grain pores, and the release of oak wood compounds including volatile and nonvolatile. Hydrolysable tannins also named ellagitannins are nonvolatile compounds and represent the most extractible oak wood phenolic compounds, they may arise in levels up to 10% in dry weight of oak heartwood [2]. *C*‒glycosidic ETs are characterized by the open-chain form of their glucose core and the presence of a C‒C linkage between the C1 of the open-chain glucose and C2 of the galloyl derived moiety of a 2,3‒HHDP (hexahydroxydiphenoyl) or a 2,3,5‒NHTP (nonahydroxyterphenoyl) unit [3,4]. Fifty years ago, vescalagin and castalagin were the first *C*-glycosidic ellagitannins monomers isolated from *Castanea* (chestnut) and *Quercus* (oak) wood [5,6]. Later, six other *C*-glycosidic ellagitannins were isolated from the same species, the lyxose/xylose derivatives and dimeric forms. More particularly, the dimers roburins A and D; the lyxose/xylose-bearing monomers grandinin and roburin E, and the lyxose/xylose-bearing dimers roburins B and C [7]. The most abundant oak wood ellagitannins are castalagin and vescalagin, which account for between 40% and 60% weight of the ellagitannins [8,9]. Figure 1 shows the main ellagitannin structures.

Ellagitannins solubilization is enhanced by the hydroalcoholic nature of wine and happens during barrel aging [10]. Their concentration is influenced by different factors like botanical and geographical origin, grain size, and all cooperage processes used like the seasoning method, and the toasting degree during barrel manufacturing [11,12,13,14,15,16]. The toasting process during barrel manufacturing participates in burning the inner surface of barrels and induces changes in ellagitannins contents, structures, and compositions. Based on the toasting intensity, ellagitannins will be more or less degraded [17,18,19]. Previous studies, using Merlot wine that has been aged with oak wood over 12 months, evidenced that the more the oak wood is toasted, the less ellagitannins are extracted into the wine [19]. At the moment, barely one research group working with purified ellagitannins from oak wood observed that when toasting takes place castalagin and vecalagin are oxidized and reduced, respectively [20]. Thus, ellagitannin derivatives formed during toasting are not well explored, the goal of the present study is to research the unknown ellagitannin structures that appeared after toasting and to see if they are formed in commercial oak wood that has been toasted.

## 2. Materials and Methods

### 2.1. Ellagitannins Extraction and Prepurification

For ellagitannins extraction, a homemade fractionation protocol was used, Figure A1 (Appendix A) shows the process [21]. The first step was employed for the ellagitannins extraction and the second for their purification. For their purification, a Toyopearl TSK HW-40 (F) gel from Tosoh Corp (Griesheim, Germany) and a C-18 column were employed. Every time, the fraction of interest was evaporated and freeze dried. The last methanol/water formic acid fraction (5/94.9/0.1, *v*/*v*/*v*) contained the 8 principal ellagitannins. The proportion of purification was calculated after both total ellagitannin and individual ellagitannin composition.

### 2.2. Estimation of Total Ellagitannin Concentration

Determination of total ellagitannins was realized as previously mentioned [16]. Every sample was analyzed in triplicate, and every reaction mixture was subjected to HPLC-UV using a Lichrospher 100 RP 18 column, 250 × 4.6 mm, 5 μm. The apparatus used for the HPLC analysis was composed of a Finnigan Surveyor UV–Vis detector (UV–Vis 200), a Finigan autosampler, and a Finnigan ternary pump. The mobile phases employed were solvent A (water/formic acid (99.9/0.1)) and solvent B (methanol/formic acid (99.9/0.1)) as well as the gradient elution was 0–35% of B in 5 min, 35–45% of B in 25 min, and 45–100% of B in 5 min. The flow rate was established at 1 mL/min with detection set at 370 and 280 nm.

### 2.3. Determination of Individual Ellagitannin Composition

The individual ellagitannin composition was also determined. The equipment used for this analysis was a Thermo-Finnigan Surveyor HPLC system composed of a Surveyor PDA plus detector, a Surveyor autosampler, and Surveyor a quaternary pump controlled by Xcalibur data treatment system. This HPLC system was also coupled to a Thermo-Finnigan LCQ Advantage spectrometer equipped with an ion trap mass analyzer. The electrospray ionization mass spectrometry detection was realized in negative ion mode with the below optimized parameters: capillary temperature 400 °C, capillary voltage 3 V, nebulizer gas flow 1.75 L/min, desolvation gas flow 1 L/min, and spray voltage 5 kV. The analyses were accomplished in duplicate on a 250 × 4.6 mm, 5 μm Lichrospher 100 RP 18 column. The mobile phases applied were solvent A [water/formic acid (99.6/0.4)] and solvent B [methanol/formic acid (99.6/0.4)]. The gradient elution was 0–3% B in 5 min, 3–12% from 5 to 35 min, and 12–100% from 35 to 40 min with a flow rate fixed at 1 mL/min and a detection wavelength maintained at 280 nm. Each ellagitannin was quantified via its molecular ion, based on external calibration curve of castalagin standard and the results are expressed as equivalents of castalagin.

### 2.4. Thermal Treatment of Ellagitannins

The above fraction obtained enclosing the eight main ellagitannins was dry heated in a laboratory oven for 60, 120, and 180 min at 220 °C. Neither the humidity nor the oxygen were controlled. After cooling, it was additionally fractionated on a C-18 column before being injected in UPLC-HRMS (High Resolution Mass Spectrometry).

### 2.5. Liquid Chromatography

The UPLC platform Agilent 1290 Infinity included a binary pumping system (1290 Infinity), a compartment thermostat column (1290 Infinity) autosampler (1290 Infinity) coupled with Diode-Array Detector (1290 Infinity). An Eclipse Plus C18 column (2.1 × 100 mm, 1.8 µm) was employed for the chromatographic separation. UV detection was carried out at 220 and 270 nm. The mobile phases employed were solvent A (H_2_O) and B (MeOH), the rate of flow was 300 μL/min, and eluent B varied as follows: 0 min,1%; 3 min, 5%; 7 min, 8%; 25 min, 12.5%; 30 min, 15%; 35 min, 20%; 40 min, 99%; 42 min, 99%; 46 min, 1%. The injection amount was 20 μL. This UPLC was connected with an HRMS, described below.

### 2.6. High Resolution Mass Spectrometry (HRMS)

UPLC was connected with an ESI-Q-TOF-MS (Agilent 6530 Accurate Mass). Mass acquisitions were realized for 42 min in negative HRMS ionization mode at 3 kV. The vaporizer temperature of the source was established at 320 °C, the capillary temperature at 350 °C, the nitrogen gas at 300 °C with a 9 L/min flow, the nitrogen sheath gas at 350 °C with a flow rate of 11 L/min, the auxiliary gas at 18, and the sweep gas at 0 (arbitrary units). The capillary voltage and the skimmer voltage were set at 3500 and 65 V, respectively. A mass range of 100–3000 Th was acquired in full scan MS mode with a mass resolution of 13,446 (*m*/*m*, fwhm at *m*/*z* 1333.968 Th). For a simple identification the fragmentor voltage was fixed at 150 V and the collision energy used for MS2 varied between 10% and 60% according to the compounds. The data obtained were analyzed by MassHunter Qualitative Analysis.

### 2.7. NMR Analysis

All 1D and 2D NMR experiments were carried out with a Bruker Avance 600 MHz NMR spectrometer (Bruker, Wissembourg, France) operating at 600.3 MHz and equipped with a 5 mm TXI probe. Data were processed using TOPSPIN software version 3.2 (Bruker Biospin, Billerica, MA, USA). All NMR spectra were acquired in methanol-d4/D2O 1:1 (*v*/*v*). Sodium [3-methylsilyl2,2′,3,3′-2H4] propionate (TSP-d4) served as an internal reference for proton chemical shifts. NMR experiments were recorded at 293 K. Molecule assignments were achieved by two-dimensional 1H-1H COSY, 1H-1H TOCSY, 1H-1H ROESY, 1H-13C HSQC, and 1H-13C HMBC experiments. All 2D experiments were carried out with 2048 data points × 400 increments and a spectral width of 8417 Hz and 33,209 Hz in proton and carbon dimension, respectively, and 1.5 s for relaxation delay. Mixing time was 300 ms and the spinlock time was 100 ms for ROESY and TOCSY experiments, respectively.

### 2.8. Isolation of Ellagitannin Thermal Products

Semipreparative HPLC was realized with an Agilent system having a quaternary pump (1260 Infinity), a compartment thermostat column (1290 Infinity), a sample injector (1260 Standard Autosampler), and a diode array detector (1260 DAD VL+). Chromatographic separation was carried out on a Prontosil column C18 column (250 × 8 mm, 5 µm, Metrohm France, Paris, France). UV detection was set at 280 nm. The mobile phases were A (H_2_O) and B (MeOH), each solvent containing 0.025% of TFA. The flow rate was set at 2.5 mL/min, and eluent B varied as follows: 1–3% during 25 min; 3–100% during 10 min, 100% during 5 min, 100 to 0% during 2 min, and 0% during 2 min. The injection volume was 200 μL. The data obtained were analyzed by LC Open Lab. Once ellagitannins compounds were isolated, they were characterized by LC-MS/MS and 1D/2D-NMR spectroscopy.

### 2.9. Wood Origin and Drying Conditions

*Quercus robur* and *Quercus petraea* were the species used for the experimentation, they originated from the Center of France. The wood pieces submitted a natural seasoning for 2 years in Nadalié cooperage seasoning park. Afterwards, different toasting temperatures were applied according to the desired final product. The intensity levels used were as follows: light (LT), 1 h and 30 min at 165 °C; light plus (LT+), 2 h at 170 °C; medium (MT), 2 h at 180 °C; medium plus (MT+), 2 h and 25 min at 190 °C; Special 2 h at 180 °C and heavy toast (HT), 2 h and 30 min at 200 °C. In the case of Special, a watering process happened just 30 min before the end of the toasting process. Two types of oak wood pieces were used, the oak chips (3–25 mm) and thin oak wood pieces (2–4 mm). Untoasted (UN) and fresh oak wood pieces were also employed for the experiment. UN wood pieces were air-dried seasoned in the seasoning park.

### 2.10. Sensory Analysis

The sensory evaluation was organized in standardized rooms (ISO-8589, 1988). For the tasting sessions, ISO black wine tasting glasses (ISO-3591, 1997) were used. To initiate the judges with bitterness taste and astringency sensation, a training was performed as previously described [19]. The final panel consisted of 19 expert judges (11 women and eight men). Everybody had either an oenology degree or an equivalent one or was working at the Institut of Vine and Wine Sciences of the University of Bordeaux (France) and agreed to take part in the sensory tests. In order to determine the tasting concentration of these two compounds, 20 wines aged in barrels representing different toastings were used and the mean concentration was 1.6 and 1.8 mg/L for compounds A and B, respectively.

To explore the incidence of the new identified compounds discrimination tests were used. More particularly a triangle test (ISO-4120, 2004) followed by a bilateral paired comparison test (ISO-5495, 2005) were realized. The objective of the first test was to determine whether each new compound had an influence on astringency sensation and bitterness taste. For each new compound, three glasses were presented, one different and two alike. Six possible order combinations were presented at random. Thus, for samples A and B, the six possible order combinations were as follows: AAB, ABA, BAA, BBA, BAB, and ABB. In total, 1.6 and 1.8 mg/L of compounds A and B, respectively, were tested individually in aqueous solution and in model wine solution for the taste/sensation identification. The panel was asked to observe the sample that was perceived different from the others. In case that a difference was found, they were allowed to describe the perceived difference. At the end of this test, bilateral paired comparison test for each compound took place, two samples were presented, and they were asked to determine between two samples which was more astringent and bitter.

Four sessions were realized, the first two using an aqueous solution as a matrix and the last two using a model wine solution. Model wine solution had the following composition: deionized water, ethanol (12%), tartaric acid (5 g/L) with pH adjusted at 3.2 with NaOH.

### 2.11. Statistical Analysis

Sensory tests results were interpreted using the theory of probability, the correct number of answers acts in accordance with a binomial distribution (*n*, *p* = 1/3 for triangle test, and *p* = 1/2 for bilateral paired comparison test), where n is the panel size. Samples were regarded as differently perceived for a probability lower than 5%.

## 3. Results and Discussion

### 3.1. Ellagitannin Purification

Ellagitannins purity was monitored during all the procedures of fractionation, via ellagic acid and individual ellagitannins quantification. Figure A2 (Appendix A) shows the purification process as well as the unique presence of individual ellagitannins in the last step of fractionation (93% purity).

### 3.2. Ellagitannins Thermal Treatment

The above purified ellagitannin fraction obtained was dry heated in an oven of laboratory for 60, 120, and 180 min at 220 °C. Once the reaction mixture was dissolved in water, the product profiles of the three reaction mixtures were analyzed by UPLC-HRMS (High Resolution Mass Spectrometry). Figure A3 (Appendix A) depicts their UPLC profile before and after toasting. It indicates that toasting creates new compounds and their intensity seems to be more important after 1 h of toasting. Thermal treatment of the eight ellagitannins generated more dehydrocastalagin, the main reaction product as well as deoxyvescalagin coeluting with vescalagin and their derivatives previously reported [20]. Besides these compounds, other ellagitannin derivatives produced by toasting exhibited [M−H]^−^ ion peaks at *m*/*z* 1055.0657 (A) and 1011.0756 (B) were detected for the first time along with a small “hill” of polymeric material (Figure 2).

### 3.3. Molecular Identification of the Isolated Compounds Using HRMS and NMR

To unambiguously identify the structure of the above A and B compounds, HRMS and NMR experiments were used. For this study, the illustration of compound A is reported to demonstrate the complementarity of these analytical techniques for determining the identity of unknown molecules. The mass accuracy, stability, and resolution supplied by HRMS are notably helpful for the experimental determination of unknown compounds formulas.

The full scan spectrum of compound A exhibits [M−H]^−^ ion at *m*/*z* 1055.0598, Chemical Formula: C_47_H_28_O_29_ as the base peak with a mass accuracy 0.21 ppm. To estimate the corresponding formula, only elements C, H, and O were considered with the following constraints: 3 < C < 150, 0 < H < 120, and 0 < O < 60. According to the literature this compound that has never been identified in oak wood but only in Japanese chestnut may correspond to Castacrenin E [22]. Since its fragmentation pattern remains unknown, fragmentation in negative mode was realized and the major fragments of this compound exhibited ions at *m*/*z* 1011.0743 (loss of CO_2_), 753.0586 (loss of an ellagic acid unit), and 300.9990 (ellagic acid), in negative mode. All these fragments correspond to ellagitannin family (Figure 3).

1D- and 2D-NMR spectra indicated the presence of an open-chain glucose. In addition to these signals a number of aromatic and ester carbons, indicating that this compound is a *C*-glycosidic ellagitannin (Table 1). In addition to these signals, the presence of an aliphatic proton singlet δ_H_ 4.66 (Cp-1) in the ^1^H-NMR spectrum and a carbonyl δ_C_ 196.9 (Cp-4), an olefinic δ_C_ 138.4 (Cp-2) and two aliphatic carbon signals δ_C_ 48.8 (Cp-1) and 90.9 (Cp-5) in the ^13^C-NMR spectra were similar to those due to the cyclopentenone ring of Mongolicain A, which is complex tannin, isolated from *Querçus species* [23]. The presence of the cyclopentenone structure was confirmed by observation HMBC correlations. The aliphatic proton at δ_H_ 4.66 (Cp-4) was correlated with olefinic carbon at δ_C_ 138.4 (Cp-2) and a carboxyl carbon δ_C_ 170.3, which was correlated with the H-2 proton of the open-chain glucose δ_H_ 5.41. In addition, the HMBC spectrum showed three-bond long-range couplings between the carbonyl carbon at δ_C_ 196.9 (Cp-4) and the H-1 of the glucose δ_H_ 4.15 and between the oxygen-bearing aliphatic carbon at δ 90.9 (Cp-5) and the H-2 of the glucose δ_H_ 5.40. The correlation pattern confirms that compound A is Castacrenin E, found previously only in Japanese chestnut, this is the first time that it has been identified in oak wood (Table 1).

For compound B, the HRMS analyses revealed the presence of [M−H]^−^ ion at *m*/*z* 1011.0669, the neutral formula C_46_H_28_O_27_ was calculated with a mass accuracy 0.35 ppm. Two mass fragments were detected: one at *m*/*z* 967.094 (loss of CO_2_), and the second at *m*/*z* 300.999 (ellagic acid). To identify this compound 1D- and 2D-NMR experiments were carried out. The NMR spectra revealed the presence of open-chain glucose pattern similar to compound A, indicating that this compound is also a *C*-glycosidic ellagitannin. Due to the structural complexity of the compound, purity and solubility problems in solvents such as deuterated water, acetone, ethanol, methanol, dimethylsulfoxide, chloroform, or mixture of these solvents, more sophisticated chemical analyses were performed. To finalize the structure determination, methylation of compound B was realized following the protocol of Tanaka et al. [22]. Unfortunately, purity and solubility problems precluded the complete elucidation of the structure of compound B. To complete the identification of this compound more complex chemical analysis will be necessary including chemical transformations or degradations. These chemical reactions require a large amount of pure compound. Currently, the purification process is time consuming and complex to obtain small amounts of pure ellagitanin. Other preparative chemical approaches need to be developed to complete the identification of compound B and other complex products extracted from oak during wood toasting.

### 3.4. Castacrenin E Origin

Castacrenin E was probably generated by oxidation of Castacrenin D. Castacrenin D is a vescalagin with an additional aromating ring (galic acid) to the C-1 through a C-C bond (Figure 4). During toasting the pyrogallol ring of Castacrenin D is replaced by a cyclopentenone ring. The oxidation pattern of this compound is similar to mongolicain A and B that are the oxidative metabolites derived from acutissimin A and B, respectively [23].

### 3.5. Identification of Ellagitannin Thermal Products on Oak Wood

The compounds A and B found in this experimental study were further searched in commercial oak wood expressing various toasting methods and sizes. Their levels were dependent on oak wood size and toasting intensity (Figure 5). Dehydrocastalagin, previously identified under experimental conditions as the toasting product of castalagin, has never been identified on commercial oak wood. As it is shown in Figure 5, independent of the size of oak wood pieces this compound even if in laboratory experiments was created from the toasting according to Glabasnia et al. [20], in real conditions is not created during toasting but it is presented naturally in commercial fresh oak wood and its concentration reached a maximum at 165 °C. Once this temperature increased to 190 °C dehydrocastalagin faded away. Deoxyvescalagin is created only by toasting and appeared only in the thin oak wood pieces when the applied temperature corresponds to a medium toast. Regarding the new identified compounds, they were created after the drying process and their concentration reached a maximum with the toasting at the temperature of the light toasting level (165 °C). The wood pieces with the smallest size presented almost two times more compounds A (Castacrenin E) and B.

More precisely, in the larger oak wood pieces Castacrenin E varies between 0.40 and 1.62 mg/g of oak wood for untoasted and light toasted oak chips, respectively. The B compound is formed only with the toasting and its concentration stays around 1.50 mg/g for both light toastings in the oak chips. In the thinner oak wood pieces, Castacrenin E ranges from 0.48 to 2.15 mg/g for untoasted and light toasted modalities. Compound B exhibits its largest concentration (2.65 mg/g) in light toasted pieces whereas in untoasted presents its minimum (0.43 mg/g) and its intermediate concentration appears in medium toasted (1.38 mg/g) thin oak wood pieces. Moreover, the compound B is the only compound to be present in medium toasting temperatures of the smallest wood pieces.

### 3.6. Sensory Impact of the New Identified Compounds

Before the sensory evaluation of the new identified compounds their purity was controlled by HRMS as well as ^1^H NMR spectroscopy; 1.6 and 1.8 mg/L of Castacrenin E and compound B, respectively, were tested in aqueous solution and in model wine solution for the taste/sensation identification. From the triangle test, 13 judges found significant differences in aqueous solution with the addition of Castacrenin E (*p* ≤ 0.01), but no significant differences were found when this compound was added in model wine solution (*p* ≥ 0.05). On the contrary, significant differences were perceived by 12 judges with the addition of compound B in model wine solution (*p* ≤ 0.01). When the compound B was added in aqueous solution no significant difference was noticed by the panel neither for astringency nor bitterness taste (*p* ≥ 0.05) (Table A1, Appendix B). According to bilateral paired comparison test, the aqueous solution with Castacrenin E was characterized more astringent and bitter than the aqueous solution alone, whereas the model wine solution with the compound B was identified as less bitter than the model wine solution alone (*p* ≤ 0.05). The results obtained indicate the importance of matrix for the sensation of astringency and bitterness. Previous studies have shown that ethanol decreases astringency sensation [24] because of ethanol intervention with hydrophobic interactions between proteins and tannins [25]. However, the taste of bitterness of some phenolic compounds is enhanced by ethanol content [24] and ethanol is indirectly implied in white wine perceived bitterness [26]. Suggesting, for both compounds, other matrixes like white and red wine should be used to fully understand their implication in astringency and bitterness taste.

Nevertheless, these findings give the first insight for the organoleptic impact of the thermal ellagitannins derivatives. Glabasnia et al. [20] noticed that ellagitannins (hydrolysable tannins) give an astringency sensation and a bitterness taste via the half-mouth test in bottled water (pH 4.5). During the half-tongue test a drop of the solution containing the compound of interest is placed on one side of the tongue when pure water is put to the other side of the tongue like conrol. After, the judges should turn their tongue in their palate during 15 s to realize if there is a difference in sensation/taste between the two sides of the tongue. From an oenological point of view, one of the major drawbacks of this test is the lack of connection between the ellagitannins and the entire oral cavity given that astringency is a sensation that can be created on non-gustatory tissues like the upper lip and gum [27]. Our observations seem to be the first ‘oenological proofs’ that ellagitannins influence directly astringency and bitterness. However, further descriptive sensory analysis needs to be done in order to see clearly how the intensity of astringency and bitterness are influenced.

Thus, this study, dedicated to the fractionation and identification of new ellagitannins formed upon cooperage toasting processes, give a first insight into the creation of two new ellagitannin compounds associated to thermal treatment with a sensory impact on astringency and bitterness. As the identification of compound B has not been completed and as one of the limits of the developed purification method is that it is time consuming, other preparative chemical approaches should be established to finalize the compound B identification. Moreover, further sensory evaluations should be performed in order to appreciate deeper the sensory impact of these new compounds.

## 4. Conclusions

Two ellagitannin derivatives which showed [M−H]^−^ ion peak at *m*/*z* 1055.0631 (compound A) and 1011.0756 (compound B) were generated by toasting and identified for the first time in French oak wood. The compound A was identified as Castacrenin E whereas the identification of compound B was not entirely achieved. The levels of these compounds in commercial oak wood were dependent on toasting intensity and varied from 0.83 to 1.68 mg/g and from 1.05 to 1.94 mg/g for the compounds A and B, respectively. Dehydrocastalagin, the toasting product of castalagin, has never been identified on commercial oak wood. Independent of the size of oak wood pieces this compound is not created during toasting it is presented naturally in commercial fresh oak wood and its concentration reached a maximum at 165 °C. Once this temperature increased to 190 °C, dehydrocastalagin disappears. Deoxyvescalagin is created only by toasting and appeared only in the thin oak wood pieces when the applied temperature corresponds to a medium toast. Regarding the new identified compounds, they were created after the drying process and their concentration reached a maximum with the toasting at the temperature of the light toast (165 °C). The wood pieces with the smallest size present almost two times more compounds A (Castacrenin E) and B. Moreover, the compound B was the only compound to be present in medium toasting temperatures. Both new compounds affected astringency sensation and bitterness taste. Further sensory analysis is needed with a view to understand profoundly the organoleptic impact of these new compounds.

## Figures and Tables

**Figure 1 foods-09-01477-f001:**
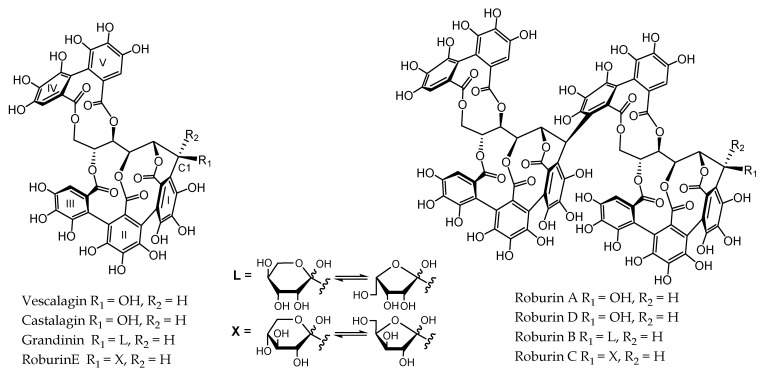
Ellagitannin structures.

**Figure 2 foods-09-01477-f002:**
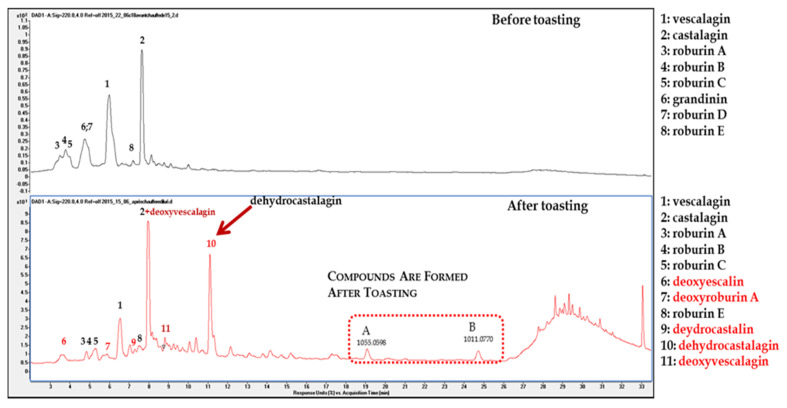
Comparative UV chromatogram before and after toasting.

**Figure 3 foods-09-01477-f003:**
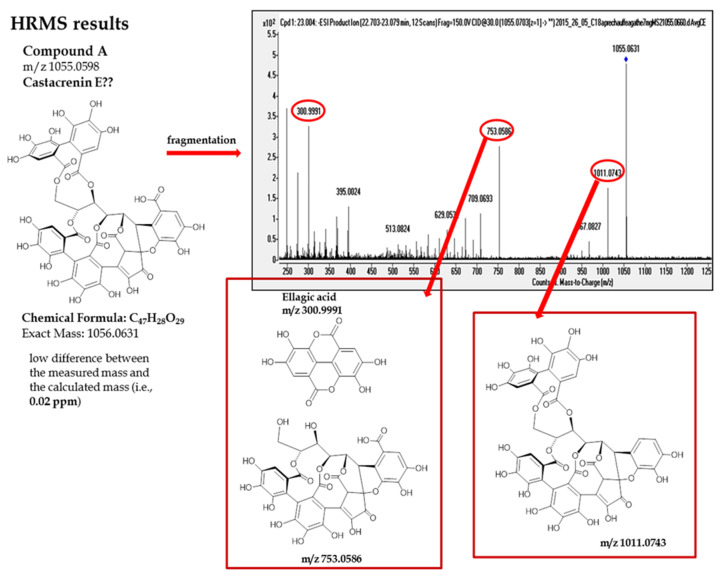
Major fragments of compound A.

**Figure 4 foods-09-01477-f004:**
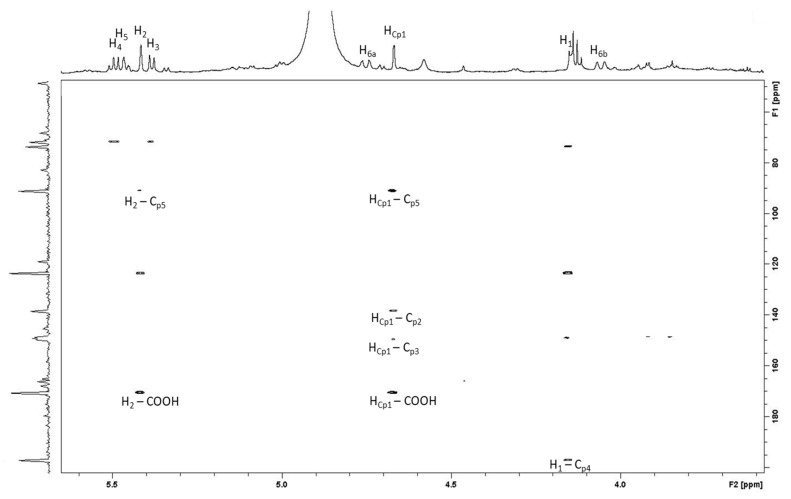
Zoom of the 1H-13C HMBC spectra in which the main correlations are indicated.

**Figure 5 foods-09-01477-f005:**
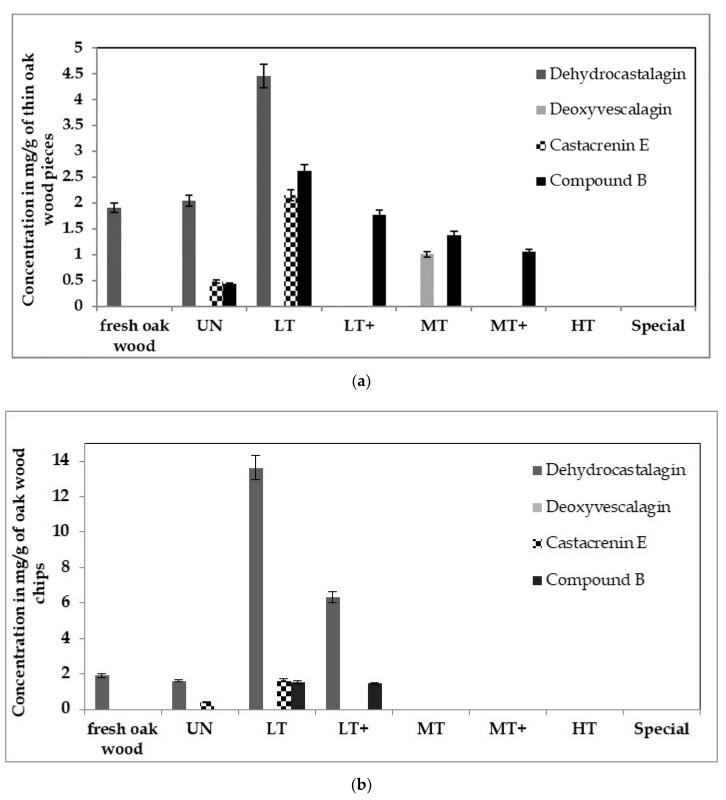
Concentration in mg/g of oak wood thin pieces (**a**), and chips (**b**). Untoasted (UN), light (LT), light plus (LT+), medium toasted (MT), medium plus (MT+), heavy toast (HT).

**Table 1 foods-09-01477-t001:** 1H- and 13C-NMR (Carbon-13 Nuclear Magnetic Resonance assignments) for Castacrenin E.

Position	δ_C_ Type	δ_H_ (Mult, *J* in Hz)
1	48.8	4.15 (s)
2	82.9	5.41 (brs)
3	73.9	5.38 (d, 8 Hz)
4	68.4	5.48 (t, 8 Hz)
5	72.1	5.45 (d, 8 Hz)
6	66.2	H6a 4.73 (dd, 2, 13 Hz)H6b 4.06 (d, 13 Hz)
Cp-1	48.8	4.66 (s)
Cp-2	138.4	
Cp-3	149.3	
Cp-4	196.9	
Cp-5	90.9	
Arom	109.3	6.35 (s)
	108.1	6.66 (s)
	108.8	6.72 (s)
	113.7	7.30 (s)
COO	167.6	
	168.1	
	168.5	
	170.7	
	170.9

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
