# Peer review of "New C-Glycosidic Ellagitannins Formed upon Oak Wood Toasting, Identification and Sensory Evaluation"

_foods, 2020, doi:10.3390/foods9101477_

Round 1

Reviewer 1 Report

The manuscript deals with analytical chemistry devoted to the fractionation and identification of novel ellagitannins that are formed upon cooperage toasting processes for wood ageing applications in alcoholic beverages and potential impacts on sensory perception.

Authors found two novel molecular features associated to thermal treatment applied to wood pieces and chips from a in-house-dry heater and from an experimental cooperage company. They monitor their presence in function of the different toasting processes. Finally, authors realize a sensory test to evaluate the astringency/bitterness perception of these two novel ellagitannins in aqueous and hydro-alcoholic solutions.

First of all, the topic is really interesting for various food applications containing wood ageing in their elaboration step since it participates to describe more precisely their chemical composition. Unfortunately, we have no information if compounds A and B are present on wood aged beverages such as wines. Nevertheless, the analytical developments were carefully carried and could be used by the scientific community to deepen the identification of novel compounds or interest in complex matrices.

However, in state, the paper require main changes prior to be published and it should be proof-read by a native English speaker. Some suggestions are listed below for each part:

Abstract :

  1. 17 dimensions

L.19 present

Introduction :

L.224-26 : add reference for all botanical wood species used for wine ageing;

L.26: “is imposed” strange

L.28: “is experienced” strange

L.32 : composed of

L.47 : used in cooperage

L.49 : manufacturing participate to burn

L.52 : been aged

MS :

L.62 : mixing

L.65-L.82 : is it a home-made fractionation protocol or is it adapted from a previous literature. Please precise and it could be nice to have a scheme of the protocol for better understanding.

L.89 and 101-102 : use rather water, methanol and formic acid to homogeneize with part 2.1

L.108 : precise please the dry heater if humidity and oxygen is controlled or not.

  1. 136 : use K rather thatn °K

L.154 : from which forest originated wood samples, if we know, pleas precise.

L.157-160 : replace OC by °C

L.162-177 : the sensory analysis is not clear. What is present exactly in the glass : the in-house thermal treated fractions or those from Nadalie? Precise their amount in the glass if possible.

Was the sensory panel trained for astringency and bitterness taste, if yes precise how.

L.175 : aqueous solution

Results :

  1. 186 : “solitary”??

L.197 : UPLC-UV is coupled to UPLC-HRMS, not clear in MS. If not it's strange to advance these information as we only see it in the next part 3.3.

Figure 2b : compouds 1 from 11 were listed based on commercial standards or based on MS spectra.

L.208 withdraw “a”

L.210 + L.236 : why authors choose the molecular assignments for peak A only with elements C,H and O ? the same for peak B ?

For peak A, MS and NMR HMBC correlation seems unambiguous for identification.

I was wondering what about adding element N in molecular assignment, does it help, particularly for peak B? Are some molecular features with sufficient matching records be plausible?

  1. 244 : purity

L.251-257 : not the same format, add a reference for the proposed mechanism.

L.263 : previously.

  1. 269 : “is disappeared” strange
  2. 293 : aqueous
  3. 327 : present

Figure 5 : precise difference between fresh wood and UN

Precise in MS the quantification of these compounds.

Were the presented neo-formed with toasting practices already mentioned in wood aged wines, please precise in text.

L.291-311 : This part is of interest and should be improved for better visibility.

is it possible to have a table with compounds that have been sensorially tested with significant or not result in aqueous and in model wine in order to visualize the obtained results.

Were the compounds tested alone or also added together (L.296 : written “ the addition of castacrenin E and Compound B), if yes could you precise in MS. Precise the concentrations tested.

Author Response

Response to Reviewer 1 Comments

  1. Include a discussion section to summarize and synthesis all the components of the study. Identify study limitations and future research that clearly emanates from your findings.

Actually the section results includes also a discussion added, the word discussion is added in L174,

In order to summarise and synthesis all the compnents of the study as well study limitations and future researh a paragraph is added  in L307. Thus this study, dedicated to the fractionation and identification of new ellagitannins formed upon cooperage toasting processes, give a first insight into the creation of two new ellagitannin compounds associated to thermal treatment with a sensory impact on astringency and bitterness. Like the complete structure elucidation of compound B has not been achieved and as one of the limits of the developed purification method is that it is time consuming, other preparative chemical approaches should be established to finalize the compound B identification. Moreover, further descriptive sensory evaluations should be performed in order to appreciate deeper the sensory impact of these new compounds.

  1. Currently, the description of the sensory methods (triangle test; identification of different sample) and the results of the triangle test (perception and influence of specific chemical compounds on perceived wine quality) do not align. Clearly identify how the specific results described were achieved by the study methods, i.e. in addition to the triangle test,

were other questions asked of the panelists.

Actually within this study we wanted to see if these compounds have an impact or not on astringency and bitterness, for this reason only triangular tests were used. In the future, when we will achieve to purify almost 1 gr of these compounds other descriptive sensory evaluations should be performed in order to appreciate deeper the sensory impact of these new compounds

  1. Improve the English language communication of the manuscript. Include

all relevant adverbs and pronouns.

The improvement is realised including relevant adverbs and pronouns

RESPONSES TO THE REVIEWER’S COMMENTS

Abstract :

  1. 17 dimensions :

Thank you for this remark, the correction is made as you suggested

L.19 present :

Thank you for this remark, the correction is made as you suggested

Introduction :

L.224-26 : add reference for all botanical wood species used for wine ageing;

Actually  it is the reference 1

L.26: “is imposed” strange : Thank your for this remark , it is replaced by introduced

L.28: “is experienced” strange : Thank your for this remark ,  it is replaced by undegone

L.32 : composed of : The correction is made as you suggested

L.47 : used in cooperage : The correction is made as you suggested

L.49 : manufacturing participate to burn : The correction is made as you suggested

L.52 : been aged : The correction is made as you suggested

MS :

L.62 : mixing : The correction is made as you suggested

L.65-L.82 : is it a home-made fractionation protocol or is it adapted from a previous

literature. Please precise and it could be nice to have a scheme of the protocol for better

understanding. 

 Yes indeed, it is a home-made fractionation protocol, a scheme is added in Appendix, L61 For ellagitannins extraction, a homemade-fractionation protocol is used, Figure A1 (Appendix 1) shows the process……

L.89 and 101-102 : use rather water, methanol and formic acid to homogeneize with

part 2.1 : The correction is made as you suggested

L.108 : precise please the dry heater if humidity and oxygen is controlled or not :

Neither the humidity nor the oxygen have been controlled….. is added as you asked

  1. 136 : use K rather thatn °K : The correction is made as you suggested

L.154 : from which forest originated wood samples, if we know, pleas precise.

The only information given by the cooperage is that the wood is coming from the forests in the Centre of France.

L.157-160 : replace OC by °C : The correction is made as you suggested

L.162-177 : the sensory analysis is not clear. What is present exactly in the glass : the inhouse thermal treated fractions or those from Nadalie? Precise their amount in the glassif possible.  

You are right, the concentration presented is not indicated, for that a phrase is added….. 1.6 mg and 1.8 mg of compounds A and B respectively were tested in aqueous solution and in model wine solution for the taste/sensation identification of bitterness and astringency.

Was the sensory panel trained for astringency and bitterness taste, if yes precise how.

Yes the panel was trained according to my previous protocol, a phrase is added ….To familiarize the panel with bitterness taste and astringency sensation, a training was performed as previously described [19]

L.175 : aqueous solution : the correction is made as you suggested

Results :

  1. 186 : “solitary”??: the word solitary is replaced by unique

L.197 : UPLC-UV is coupled to UPLC-HRMS, not clear in MS. If not it's strange to advance

these information as we only see it in the next part 3.3.

A phrase is added in L 95, This UPLC was connected with an HRMS, described below.

Figure 2b : compouds 1 from 11 were listed based on commercial standards or based on

MS spectra.

Compounds were listed based on MS spectra

L.208 withdraw “a” : the correction is made as you suggested

L.210 + L.236 : why authors choose the molecular assignments for peak A only with

elements C,H and O ? the same for peak B ?

We show you the molecular formulas with only the selected elements C, H and O because it was the higher score and matching formula between the measured mass and the calculated mass. All the proposed formulas with N, or halogen atoms exhibits very low score and high ppm differences between measured and calculated mass. The same for peak B.

For peak A, MS and NMR HMBC correlation seems unambiguous for identification.

I was wondering what about adding element N in molecular assignment, does it help,

particularly for peak B? Are some molecular features with sufficient matching records

be plausible?

Actually the identification was achieved based on the reference 21. Tanaka, T.; Ueda, N.; Shinohara, H.; Nonaka, G. I.; Kouno, I., Chem. Pharm. Bull. 1997.

As previously mentioned we used C, H and O because it was the higher score and matching formula between the measured mass and the calculated mass.

  1. 244 : purity, The correction is made as you suggested

L.251-257 : not the same format, add a reference for the proposed mechanism. The correction is made as you suggested, the reference 22 is added

L.263 : previously. The correction is made as you suggested

  1. 269 : “is disappeared” strange, The word dissapeared is replaced by faded away
  2. 293 : aqueous . The correction is made as you suggested
  3. 327 : present . The correction is made as you suggested

Figure 5 : precise difference between fresh wood and UN : the difference is that the UN have not been air dried seasoned. A phrase has been added in L149. Untoasted (UN) and Fresh oak wood pieces have also been employed for the experiment. UN wood pieces have been air dried seasoned in the seasoning park.

Precise in MS the quantification of these compounds.

The precision is made in L89-90, Each ellagitannin was quantified via its molecular ion, based on external calibration curve of castalagin standard and the results are expressed as equivalents of castalagin

Were the presented neo-formed with toasting practices already mentioned in wood

aged wines, please precise in text.

Actually yes but it is a part of another research project that we are working on, once the results are validated a publication will be done

L.291-311 : This part is of interest and should be improved for better visibility.

is it possible to have a table with compounds that have been sensorially tested with

significant or not result in aqueous and in model wine in order to visualize the obtained

results.

You are right, a figure is made, it is the figure 6.

Were the compounds tested alone or also added together (L.296 : written “ the addition

of castacrenin E and Compound B), if yes could you precise in MS. Precise the

concentrations tested.

The compounds were tasted individually, the next step will be to mix these compounds with other ellagitannin compounds like vescalagin and castalagin. The precision is made in MS,  1.6 mg and 1.8 mg of compounds A and B respectively were tested individually in aqueous solution and in model wine solution for the taste/sensation identification.

Reviewer 2 Report

This work presents interesting results pertaining to the oenological research. The identification of new oak wood compounds improve the knowledge of volatile aroma of aged wines. These new compounds could be used as markers for barrel-makers. The research group have used different analytical methodologies to improve the identification of the compounds. However, the most important it is the impact of these compounds on sensorial analysis.

In Materials and Methods

Line 61, please provide a reference or explain the protocol.

Line 101, 120 and 134, please correct H2O and D2O.

In Results and Discussion

Line 174, please correct the word “discusion” with discussion

Line 183-185, Could you indicate more detail about new compounds quantity?

Line 197-198, I think that it’s not necessary to cite the advantages of HMRS methodology.

Line 287, “intensified the intensity of astringency and bitterness” please, check the sentence.

Line 286-288, I have a big question about this affirmation “the addition of the compound B on model wine solution reduced bitterness intensity”. Is the model wine solution bitter? Why? It’s the same observation for Castacrenin E that intensify the astringency and bitterness of aqueous solution…the aqueous solution is astringent and bitter. I think that you have to explain these differences or express better your results. Triangle test is to differentiate among samples but not identify intensity, so you should write this paragraph in a different way or give more information.

Author Response

Line 61, please provide a reference or explain the protocol : The reference 21 is added

Line 101, 120 and 134, please correct H2O and D2O. : What I should correct, i  cannot see, can you precise me please ?

In Results and Discussion

Line 174, please correct the word “discusion” with discussion : Thank you for the remark, the correction is made

Line 183-185, Could you indicate more detail about new compounds quantity?: Actually in terms of quantity the new compounds were presented in a quantity around 3 times less than vescalagin

Line 197-198, I think that it’s not necessary to cite the advantages of HMRS methodology. We wanted only to show you the process that we used to obtain the formula for the first compound

Line 287, “intensified the intensity of astringency and bitterness” please, check the sentence. Thank you for this remark, the sentence is checked and is replaced by intensified the astringency and bitterness

Line 286-288, I have a big question about this affirmation “the addition of the compound B on model wine solution reduced bitterness intensity”. Is the model wine solution bitter? Why? It’s the same observation for Castacrenin E that intensify the astringency and bitterness of aqueous solution…the aqueous solution is astringent and bitter. I think that you have to explain these differences or express better your results. Triangle test is to differentiate among samples but not identify intensity, so you should write this paragraph in a different way or give more information.

Thank you for your remarks, Modifications have been done accordingly.

Line 286,Before the sensory evaluation of the new identified compounds their purity was controlled by HRMS as well as 1H NMR spectroscopy; 1.6 mg/L and 1.8 mg/L of Castacrenin E and compound B respectively were tested in aqueous solution and in model wine solution for the taste/sensation identification. From the triangle test, the judges found significant differences on aqueous solution with the addition of Castacrenin E (p ≤ 0.01), but no significant differences were found when this compound was added in model wine solution (p ≥ 0.05). On the contrary, significant differences were perceived with the addition of compound B in model wine solution (p ≤ 0.01). When the compound B was added in aqueous solution no significant difference was noticed by the panel neither for astringency nor bitterness taste (p ≥ 0.05) (Table A1, Appendix2). According to bilateral paired comparison test, the aqueous solution with Castacrenin E was characterized more astringent and bitter than the aqueous solution alone , whereas the model wine solution with the compound B was identified as less bitter (p ≤ 0.05) than the model wine solution alone. The results obtained indicate the importance of matrix for the sensation of astringency and bitterness. Previous studies have shown that ethanol decreases astringency sensation [24] because of ethanol intervention with hydrophobic interactions between proteins and tannins [25]. However, the taste of bitterness of some phenolic compounds is enhanced by ethanol content [24] and ethanol is indirectly implied in white wine perceived bitterness [26]. Suggesting, for both compounds, other matrixes like white and red wine should be used to fully understand their implication in astringency and bitterness taste.

Reviewer 3 Report

The research try to clarify the nature of the different compounds apported by the oak wood to wine during ageing.
For better comprehension I suggested to write in Figure 5 the name of acronyms: that is : Untoasted UN, light (LT),  light plus (LT+), medium toasted (MT), ; medium plus (MT+), 2; Noisette, Special  and heavy toast (HT), indicating in X axe "toasting grade" or similar.

I remark one results that is important enough to be considered as a conclusion:

"Our observations seem to be the first ‘oenological proofs’ that ellagitannins influence directly astringency and bitterness".

• Its relevance is medium, but it contributes to known how different ellagitannin compounds can influence wine aroma.
• The topic is original and strong in my opinion from chemical point of view. Several studies have been done to understand the sensorial nature of the compounds that are apported during ageing, and many of then has contributed to the knowledge of the complex phenomena that happens during wine ageing in oak barrel or wood. That is the case of this research.
It would be great if it would be possible identify the two components, not just one of them.
Also, and for future research I suggest using real wine instead model wines, as it is known that ellagitannins reacts one each other and the real effect in wine can be different.
Also, could be a good idea to analyzed real wine coming from barrels, to see if its possible to identify these two compounds.
• The paper well written and the text is clear. Figures are quite explicative. They help the understanding of the text.
• Conclusion are consistent and they address the main objective of the paper I remark one result that is important enough to be considered as a conclusion: "Our observations seem to be the first ‘oenological proofs’ that ellagitannins influence directly astringency and bitterness".

Author Response

Response to Reviewer 3 Comments

 The research try to clarify the nature of the different compounds apported by the oak wood to wine during ageing.
For better comprehension I suggested to write in Figure 5 the name of acronyms: that is : Untoasted UN, light (LT),  light plus (LT+), medium toasted (MT), ; medium plus (MT+), 2; Noisette, Special  and heavy toast (HT), indicating in X axe "toasting grade" or similar.

 Thank you for your remarks, the name of acronyns has been added in figure 5

I remark one results that is important enough to be considered as a conclusion:

"Our observations seem to be the first ‘oenological proofs’ that ellagitannins influence directly astringency and bitterness".

Thank you for your remarks

• Its relevance is medium, but it contributes to known how different ellagitannin compounds can influence wine aroma.
• The topic is original and strong in my opinion from chemical point of view. Several studies have been done to understand the sensorial nature of the compounds that are apported during ageing, and many of then has contributed to the knowledge of the complex phenomena that happens during wine ageing in oak barrel or wood. That is the case of this research.
It would be great if it would be possible identify the two components, not just one of them.
Also, and for future research I suggest using real wine instead model wines, as it is known that ellagitannins reacts one each other and the real effect in wine can be different.
Also, could be a good idea to analyzed real wine coming from barrels, to see if its possible to identify these two compounds.

The identification of the B compound is in progress and in the future we plan to add these compounds in a red wine. The wine coming from barrels has been already analysed and we find these compounds

• The paper well written and the text is clear. Figures are quite explicative. They help the understanding of the text.

Thank you
• Conclusion are consistent and they address the main objective of the paper I remark one result that is important enough to be considered as a conclusion: "Our observations seem to be the first ‘oenological proofs’ that ellagitannins influence directly astringency and bitterness".

Thank you for your comments

Reviewer 4 Report

The paper “New C-Glycosidic ellagitannins formed upon oak 2 wood toasting; Identification and sensory evaluation” reports the finding of one (maybe two) previously unreported ellagitannins from oak wood and describes the role of wood toasting on the degradation of some ellagitannins. The instrumental analytical approach is satisfactory and the results are convincing and fruitful to the field. However, some major issues arise from the sensory evaluation part of the research. The main concern is about the method applied to assess the astringency and bitter properties of the tested tannins. Such sensory properties have additive effect in mouth when sequential tastings are performed thus affecting the reliability of the results. As very atypical results were reported in the paper (astringency decreases in model wine solution following to addition of tannin) a detailed description of the method applied to the sensory analysis is essential.

Author Response

The paper “New C-Glycosidic ellagitannins formed upon oak 2 wood toasting; Identification and sensory evaluation” reports the finding of one (maybe two) previously unreported ellagitannins from oak wood and describes the role of wood toasting on the degradation of some ellagitannins. The instrumental analytical approach is satisfactory and the results are convincing and fruitful to the field. However, some major issues arise from the sensory evaluation part of the research. The main concern is about the method applied to assess the astringency and bitter properties of the tested tannins. Such sensory properties have additive effect in mouth when sequential tastings are performed thus affecting the reliability of the results. As very atypical results were reported in the paper (astringency decreases in model wine solution following to addition of tannin) a detailed description of the method applied to the sensory analysis is essential.

Thank you for your remarks, More details are added in the sensory analysis section as you asked. Due to saturation and persistence of the bitter taste as well as palate fatigue of the panel, panelists were asked to rinse mouth with water and wait one minute between each sample.

Line

Comment

32

Change dispose to have

The change is made

148

“…fresh oak wood pieces…”

The change is made

151

Change realized to performed

The change is made

163

Do Authors mean 1.6 mg/L and 1.8 mg/L?

Yes indeed the mg/L is added

165

Bad English “..to observe the one gustatory perceived as different..”

The English is changed… The panel was asked to observe the sample that was perceived different from the others. In case that a difference was found, they were allowed to describe the perceived difference.

Supplementary figures

Supplementary figures are badly identified. Figure A3 reports chromatograms of both differently purified fractions and differently toasted wood.

As it mentioned in L177 the purified ellagitannin fraction was dry-heated in an oven of laboratory for 60 min, 120 min and 180 min at 220°C. The goal was to see at which temperature we obtain more degradation ellagitannin products

195

“…the illustration of compound A (Figure 3)… is reported…”

The changed is made

196

What do Authors mean with the term “supplementarity”?

We wanted to say complementary

205

an ellagic acid

The change is made

245

Do Authors mean “pyrogallol ring”?

Yes thank you, the change is made

282

How the tannin amounts were chosen?

We quantify these compound in red wine aged in barrels and we choose the amount that represent the mean concentraton of 20 wines

Figure 6

The picture does not bring any significant information to the text. Remove. Instead, a table summarizing the results and statistics (water solution and model wine, bitterness and astringency) of the triangular sensory test should be included here or as supplementary table.

The change is made, the figure is removed

298

What do authors mean by writing “oenological proof”? Do they refer to the influence in synthetic wine or do they generalize? They tested two peculiar ellagitannins at peculiar amount, but they do generalize to all the ellagitannins.

No we mean in wine conditions, as up to now we had the sensory impact of ellagitannins in conditions that we do not have in the wine…we do not generalize, is only an observation and we will continue to work in ellagitannins sensory impact

The amount tested (1.6-1.8 mg/L ?) is usually very low for perceiving bitterness and astringency, especially in ethanol solutions. How do authors explain it? Actually we choose the mean concentration find in 20 wines as explained you before but of course the next step is to increase the concentration and use the maximum one find in the wines.

308

“Like the complete structure elucidation of compound B has not been achieved..” bad English. Please, revise.

The revision is made,  Like the identification of compound B has not been completed

314

Delete “To conclude”

The change is made

316

Change like to as

The change is made

318-319

Which solubilization conditions were considered for expressing data as concentration (mg/L)?

They were soluble in water

Round 2

Reviewer 1 Report

Even if it is not easy to track authors modifications in the revised version, after a careful reading of the revised article, the scientific message of the article is much clearer and it can be useful for the scientific community. The article could be published in state.

Author Response

(The authors gave the same response as above.)

Reviewer 4 Report

The most of the following comments were reported in my first revision. However, they were replied but not suitably took into account in the text, yet.

Line

Comment

Missing

Supplementary figures

Supplementary figures are badly identified. Figure A3 reports chromatograms of both differently purified fractions and differently toasted wood.

Figures title must be self-explaining even if supplementary. The fig A3 does not reports “Compounds formed after different times of toasting” but “Effect of oak wood toasting conditions on the chromatographic pattern of purified extracted tannins”

Supplementary  table A1

Did all of the 12 correct-responding judges (different sample) stated less astringent/bitter the wine added with compound B? Please specify also in the text in chapter 3.6

196

What do Authors mean with the term “supplementarity”?

Not amended in the text

283

How the tannin amounts were chosen?

Explanation reported neither in the text nor in the material and method section.

316

“Like the complete structure elucidation of compound B has not been achieved..” bad English. Please, revise.

Change “Like” to “As”.

326-328

Which solubilization conditions were considered for expressing data as concentration (mg/L)?

Content of Castacrenin E (why is it reported as compound A in line 328?) and compound B in oak wood (a solid) is reported as mg/L (as liquid).

Are the reported range values referring to the commercial samples in figure 5? If not,  which wood samples are they referring to?

Author Response

Line

Comment

Missing

Responses

Supplementary figures

Supplementary figures are badly identified. Figure A3 reports chromatograms of both differently purified fractions and differently toasted wood.

Figures title must be self-explaining even if supplementary. The fig A3 does not reports “Compounds formed after different times of toasting” but “Effect of oak wood toasting conditions on the chromatographic pattern of purified extracted tannins”

Compounds formed after different times of toasting is replaced by Effect of oak wood toasting conditions on the chromatographic pattern of purified extracted tannins in Figure A3

Supplementary  table A1

Did all of the 12 correct-responding judges (different sample) stated less astringent/bitter the wine added with compound B? Please specify also in the text in chapter 3.6

Actually for 19 judges the minimun number of correct responses to establish a significant difference is 11 for p ? 0.05 and 12 for p  ? 0.01; The number of right answers is put in the text, section 3.6.From the triangle test, thirteen judges found significant differences on aqueous solution with the addition of Castacrenin E (p ? 0.01), but no significant differences were found when this compound was added in model wine solution (p ? 0.05). On the contrary, significant differences were perceived by twelve judges with the addition of compound B in model wine solution (p ? 0.01).

196

What do Authors mean with the term “supplementarity”?

Not amended in the text

complementarity is replaced supplementarity in L196

283

How the tannin amounts were chosen?

Explanation reported neither in the text nor in the material and method section.

20 wines aged in barrels representing different toastings were used and the mean concentration was 1.6 mg/L and 1.8 mg/L for compounds A and B respectively. The phrase is added in section 2.10

316

“Like the complete structure elucidation of compound B has not been achieved..” bad English. Please, revise.

Change “Like” to “As”.

The change is made

326-328

Which solubilization conditions were considered for expressing data as concentration (mg/L)?

Content of Castacrenin E (why is it reported as compound A in line 328?) and compound B in oak wood (a solid) is reported as mg/L (as liquid).

Thank you for your remark, the unit is mg/g of oak wood

For the solubilisation water and model wine solution were used. For exemple in the case of 1,6 mg/L of Castacrenin E in aqueous solution, 1,6 mg was dissoved in 1L of aqueous water

Are the reported range values referring to the commercial samples in figure 5? If not,  which wood samples are they referring to?

Indeed the wood correspond to commercial samples